# Using natural language and program abstractions to instill human inductive biases in machines

Sreejan Kumar[1], Carlos G. Correa[1], Ishita Dasgupta[2], Raja Marjieh[3], Michael Y. Hu[4],
Robert D. Hawkins[1,3], Nathaniel D. Daw[1,3], Jonathan D. Cohen[1,3], Karthik Narasimhan[4], and
Thomas L. Griffiths[3,4]

[1]Princeton Neuroscience Institute
[2]DeepMind
[3]Department of Psychology, Princeton University
[4]Department of Computer Science, Princeton University

## Abstract

Strong inductive biases give humans the ability to quickly learn to perform a variety of tasks. Although meta-learning is a method to endow neural networks with useful inductive biases, agents trained by meta-learning may sometimes acquire very different strategies from humans. We show that co-training these agents on predicting representations from natural language task descriptions and programs induced to generate such tasks guides them toward more human-like inductive biases. Human-generated language descriptions and program induction models that add new learned primitives both contain abstract concepts that can compress description length. Co-training on these representations result in more human-like behavior in downstream meta-reinforcement learning agents than less abstract controls (synthetic language descriptions, program induction without learned primitives), suggesting that the abstraction supported by these representations is key.

## 1 Introduction

Humans are able to rapidly perform a variety of tasks without extensive experience [1]. This may be because of strong inductive biases towards abstract structured knowledge (e.g. hierarchies, compositionality) [2, 3] that act as strong prior knowledge, enabling generalization to novel environments with little new data. These biases present one of the most salient differences between humans and neural network-based learners [4] and may be one of the keys to building artificial agents with human-like generalization capabilities. For this reason, there has been interest among machine learning researchers in identifying and instilling these inductive biases into neural network-based agents [5–7].

One emerging approach to implicitly bestowing inductive biases on neural networks is *meta-learning* [8, 9]. In meta-learning paradigms, an agent is trained not just on a single task but on a *distribution* of tasks, with the aim of acquiring the underlying abstractions that these tasks have in common. However, since neural networks are not easily interpretable, it can be difficult to tell if the resulting neural networks actually acquired this abstract knowledge, or whether they have instead learned statistical artifacts correlated with abstract rules [10].

What representations can give our artificial agents access to human inductive biases? Previous work in reinforcement learning has shown that neural networks can improve performance through auxiliary tasks [11–13]. We build on this insight to construct auxiliary tasks that require reproduc-

36th Conference on Neural Information Processing Systems (NeurIPS 2022).

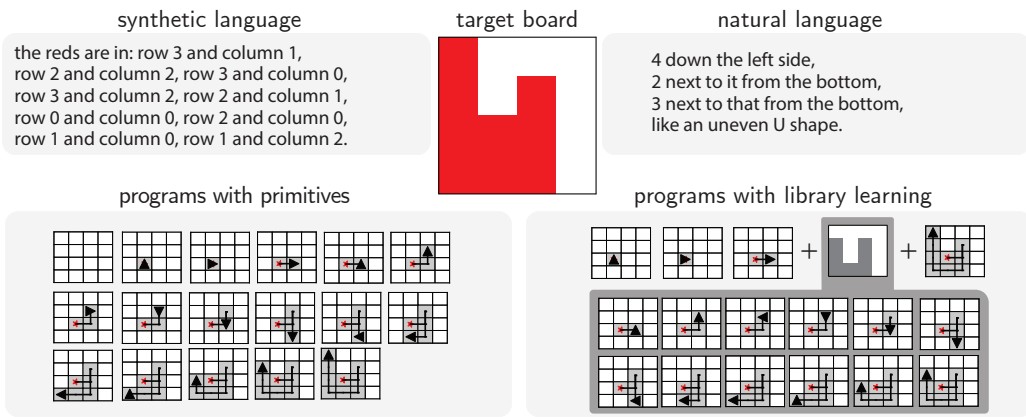

Figure 1: **Four repositories of human-like priors.** For a stimulus space of 2D binary grids, we investigate two classes of repositories for human-like priors: linguistic descriptions (top row) and program abstractions (bottom row). For linguistic descriptions, we consider both synthetic language (upper-left), and human-generated language (upper-right). For program abstractions, we implemented a simple DSL consisting of an agent that can move around on the grid while filling-in red tiles. We consider both primitive programs written in the base DSL (bottom-left) as well as programs written with a learned library of abstractions (bottom-right).

ing different kinds of representations, and examine what kinds of representations tend to produce agents with human-like inductive biases.

Human biases toward abstract knowledge might be linked to the ability to verbalize this knowledge through natural language [14, 15]. Human language descriptions can therefore act as a repository of this prior knowledge. Recent work in machine learning has shown that neural network representations can be shaped and structured through natural language supervision for various tasks [16–22]. Abstract knowledge in humans has also been modeled by *program induction* [23–26], where a model directly infers a structured programmatic representation from data. These programmatic representations are constructed from primitive atoms using symbolic rules and procedures. Recent work has leveraged neural networks to make this otherwise expensive and brittle process more scalable [*neurosymbolic models*; 27–29]. In these approaches, abstract knowledge is explicitly built into the artificial agent using a pre-specified Domain Specific Language (DSL), which can be restrictive. In many situations, we would like artificial agents that are *implicitly* guided towards human-like abstract knowledge [30–33] rather than explicitly building it in, since it can be hard to determine what specific concepts to build into a DSL for any arbitrary environment. Therefore, in this work, we focus on how to use representations from program induction to guide artificial agents that don't have these built in concepts.

In this work, we show that language and programs can be used as *repositories* of abstract prior knowledge/inductive biases of humans that one can co-train with to elicit human-like biases in other related tasks. The differences in behavior elicited by the use of different inductive biases are subtle, so we use a recent framework from Kumar et al. [10] to build tasks that distinguish these subtle differences. The specific domain we use is a meta-reinforcement learning task where the agent sequentially reveals patterns on 2D binary grids. We find that guiding meta-reinforcement learning agents with representations from language and programs not only increases performance on task distributions humans are adept at, but also decreases performance on control task distributions where humans perform poorly. We also show a correspondence between human-generated language and programs synthesized with *library learning*. This indicates that humans use higher-order concepts to compress their descriptions in the same way that recent library learning approaches in program induction add new concepts to the DSL to compress program length [24]. Finally, we show evidence that this process of *abstraction* through compression in these representations (either language or program) is a key driver of *human-like* behavior in the downstream meta-learning agent, when co-trained on these representations.

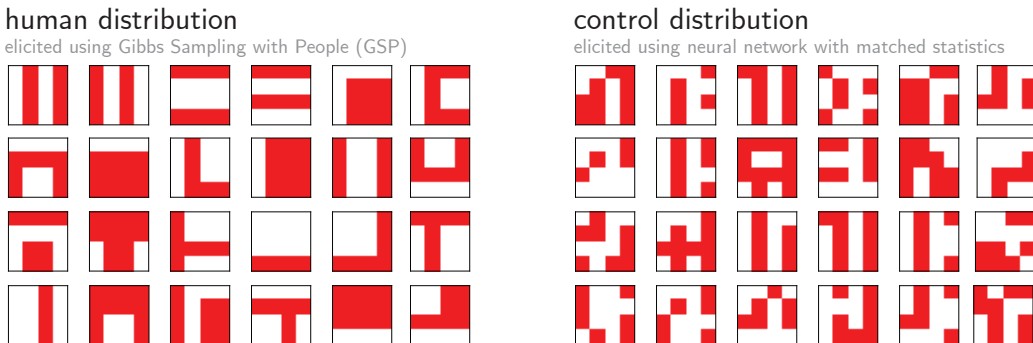

human distribution
elicited using Gibbs Sampling with People (GSP)

control distribution
elicited using neural network with matched statistics

Figure 2: Example grids from human-elicited priors (left) and machine-generated priors (right).

## 2 Dataset and task paradigm

Our goal is to examine differences in how human-like the behavior of an agent is, as elicited by different co-training objectives. To do this, we first need a distribution that reflects human inductive bias (as well as a control which is closely matched but doesn't directly reflect this inductive bias) – so that human-like behavior is distinguishable. We introduce these distributions in the following sections, as well as how we build an RL task based on these distributions.

### 2.1 Eliciting human priors using Gibbs sampling

We collected a dataset where we directly sample from humans' prior expectations on two-dimensional binary grids (we will refer to these as "boards"). These data contain the direct human prior knowledge/inductive biases and can therefore be used to study them. To do this, we used a technique called Gibbs Sampling with People [GSP; 34, see Fig. 2] that samples internal prior distributions by putting humans "in the loop" of a Gibbs sampler. The stimulus space consisted of the space of $4 \times 4$ boards giving 16 stimulus dimensions. Each stimulus dimension (corresponding to a tile on the board) had two possible values and determined the binary color of the tile, namely, red or white. Each GSP trial consisted of a human participant predicting what color a single masked square in the grid should be, conditional on the colors of all other squares on the grid. The specific instructions given were "What should be the underlying color of the covered greyed tile such that the board is generated by a very simple rule?" (Fig. 2). Once a decision was made the resulting stimulus was passed on to a new participant, who repeated the task with a different masked tile. When sampling from this distribution, the probability of each board is based on how frequently it occurred during the GSP sampling process, which reflects its probability under human priors over the space of 2D grids. We selected the $500$ most probable boards to collect language and program data. We will refer to these as the "GSP boards".

### 2.2 Constructing a matched control distribution

The control distribution of boards matches the statistics of the GSP boards but is not produced by human decisions (see Fig. 2). A fully connected neural network (3 layers, 16 units each) was trained to encode the conditional distributions of the GSP boards: a random tile is masked out, and the network is trained to predict its value given the other tiles (similar to masked language models; [35]). These conditional distributions contain all the relevant statistical information about the boards. The network achieved an accuracy of above 99% on this task.

This network is then used to generate samples, using the same process that was used to generate samples from human priors via Gibbs sampling. A board in which each tile is randomly set to red or white with probability 0.5 is initialized, and this trained network is used to predict masked out tiles. Since the conditional model is trained on the GSP boards, this generates a set of boards with similar statistical properties to the original GSP boards (example: number of red tiles in the GSP boards (Mean=8.4, SD=2.26) do not significantly differ from the control boards (Mean=7.4, SD=2.01), $p = 0.12$), but also implicitly encode the priors of the conditional model (i.e. a trained neural network instead of humans) as well.

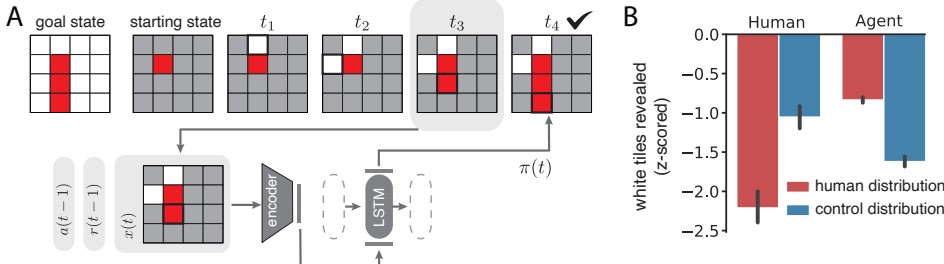

Figure 3: **Meta-RL Task and Architecture.** (A) The goal of the agent is to sequentially uncover all the red tiles to reveal a picture on a 2D grid while uncovering as few white tiles as possible. A single board is one task and a distribution over boards is a task distribution. (B). Performance of humans and artificial agents on the human-elicited vs machine-generated samples (see Fig 2). The performance metric is based on the number of white tiles revealed, so lower is better.

## 2.3 A search task for meta-reinforcement learning

**Task.** These distributions of boards can be used to construct new reinforcement learning tasks. We use the same task, performance measures, and agent architecture as in Kumar et al. [10]. In this task, the agent starts with an almost fully masked out board (all greyed tiles except for one revealed red), and has to select tiles to reveal. The revealed tiles are red or white depending on the tile colors of a preselected underlying board. One board with a fixed configuration of red tiles defines a single task. The agent's goal is to reveal all the red tiles, while revealing as few white tiles as possible. The episode ends when the agent reveals all the red tiles. There is a reward for each red tile revealed, and a penalty for each white tile revealed. Agents are trained on a set of boards and tested on held out test boards. Doing well on this task requires the agent to learn and represent the distribution over boards in its training distribution. The primary manipulation is whether a board is generated from human priors or from the control distribution (that is closely matched in terms of statistics, but generated from machine priors). Learners that do better on boards that were directly sampled from people's prior distribution on 2D grid stimuli can be said to have a human-like inductive bias, and vice versa for machine-generated boards (see Fig. 2).

**Evaluation metric.** The specific metric we use to track performance is as follows. An agent that does well on the task will reveal all the red tiles while revealing as few white tiles as possible. We therefore measure performance by counting the number of white tiles revealed in the episode. Since different boards will have different number of red tiles and thus have varying levels of difficulty, we measure the performance relative to a "nearest neighbor" heuristic. This heuristic randomly selects covered tiles adjacent to currently uncovered red tiles. We run this heuristic 1000 times on each board and the human/agent's number of white tiles revealed is z-scored according to this distribution. A z-score of below 0 means that the human/agent did better than the mean performance of the nearest neighbor heuristic. The specific value of the z-score reflects how many standard deviations the human/agent did better than the mean performance of the nearest neighbor heuristic. We determine significant differences in performance across different task distributions and models using a non-parametric bootstrap independent samples statistical test.

**Baseline experiments.** The agent architecture is an LSTM meta-learner trained with reinforcement learning. This architecture is inspired by Wang et al. [36]. Our LSTM meta-learner takes the full board $x_t$ as input, passes it through convolutional and fully connected layers and feeds that, along with the previous action $a_{t-1}$ and reward $r_{t-1}$, to 120 LSTM cells. The agent had 16 possible actions corresponding to choosing a tile (on the $4 \times 4$ board) to reveal. The reward function was: +1 for revealing red tiles, -1 for white tiles, +5 for the last red tile, and -2 for choosing an already revealed tile. The agent was trained using Proximal Policy Optimization [PPO; 37] using the Stable Baselines package [38] for one million episodes.

We trained our agent on the distribution of the human-generated GSP samples (see Fig 2). We then evaluated the agent on a set of human-generated samples as well as machine-generated samples, both of which were held out of training. Despite the fact that we did not train the agent on the machine-generated samples, the agent still performs significantly better on the control samples than the human samples ($p < 0.0001$) (Fig. 3B), which is in stark contrast to humans who perform

significantly better on the human-generated samples ($p < 0.0001$). The fact that the meta-RL agent exhibits the opposite pattern of performance across human and machine-generated tasks (compared to humans) suggests that the agent has learned a different representation than that used by the humans, reflecting a different set of inductive biases than that of the humans. This is a replication of the effect reported in Kumar et al. [10] – using the same architectures and task paradigm, but here the agent is trained with PPO instead of A2C. Having established that humans and agents seem to use different representations to perform these tasks – presumably driven by differences in inductive bias – we now consider how to provide the agent with more human-like inductive bias(es).

## 3   Instilling human inductive biases with language and programs

### 3.1   Natural language representations

We collected natural-language descriptions of 500 of the most probable GSP boards from a naive group of participants. Each participant wrote descriptions for 25 unique boards in response to the following prompt: "Your goal is to describe this pattern of red squares in words. Be as detailed as possible. Someone should be able to reproduce the entire board given your description. You may be rewarded based on how detailed your description is." Participants were randomly assigned boards such that each board had 9-12 descriptions from different participants. As a control for the language data, we also generated *synthetic* text descriptions. Synthetic descriptions used a template that said "The reds are in: row $X_1$ and column $Y_1$,...," for every red tile location $(X_n, Y_n)$. We permuted the order in which the red tile locations were verbalized to generate multiple descriptions per board and mimic the multiple human descriptions per board for the human-generated descriptions. To embed each description (both human-generated and synthetic) into a vector space with semantic meaning, we obtained the RoBERTa [39] sentence embedding for each description using the SentenceTransformer package (`https://www.sbert.net/`, based on Reimers and Gurevych [40]). The SentenceTransformers model maps text into a 768 dimensional dense semantically meaningful vector space (i.e. text samples that are semantically similar are close to each other in vector space). Reimers and Gurevych [40] do this by using a contrastive objective on a dataset of semantically-paired text where embeddings from the same pair are pushed closer and embeddings from different pairs are pushed further apart.

### 3.2   Program representations

We used DreamCoder [24] to parse the boards from the GSP experiment into programs that can generate them. Given a domain-specific language (DSL) and a way to score a program comprised of DSL functions, DreamCoder enumerates and scores programs during its *wake phase*. This process is accelerated in two ways during the *sleep phase*: first, by learning a neural-network-based *recognition model* that guides searching the space of programs during wake, and second by augmenting the DSL with subprograms abstracted from repeated expressions in existing programs using *library learning*. DreamCoder alternates between wake and sleep phases for multiple cycles.

We now describe our domain and DSL. In this domain, the goal is to produce the target board on a $4 \times 4$ grid by controlling a pointer that marks locations with a pen (see Fig. 1). The pointer state consists of a location on the grid, an orientation in one of the four cardinal directions, and whether the pointer's pen is down (i.e. whether it will mark at subsequent future locations). The DSL is comprised of the following primitives: `move` which moves the pointer forward from the current location based on orientation; `left`, `right` which turn the pointer, changing its orientation; `pen-up`, `pen-down` which change the state of the pointer's pen; and `fork` which stores the pointer state, executes a program, and then restores the agent state (except that grid locations marked in the subroutine are not reset).

The score $s(\rho|l)$ of a program $\rho$ when executed from an initial location $l$ is the sum of several penalties that encourage the program to replicate the target board faithfully in an efficient manner. First, there is a penalty of $-10$ for each marked location that is in the target board and not in the generated program's board, and a penalty of $-\infty$ for each location that is in the generated pro-

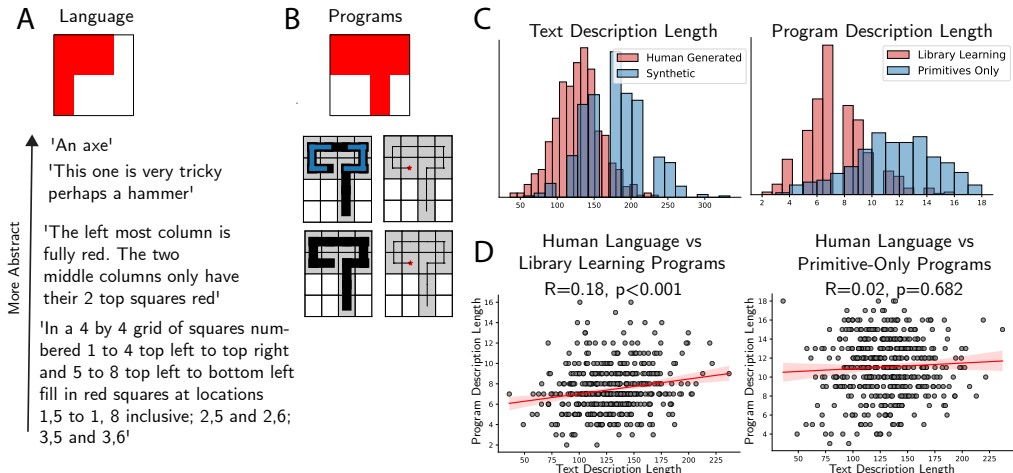

Figure 4: **Properties of Language and Program Data.** (A) Example board with corresponding human-generated language descriptions. Some people write more abstract descriptions than others. (B) Examples showcasing compression of programs with learned library concepts. We show the library-learning program (top grids) and non-library learning programs (bottom grids). For each program, we show the execution path (right, star indicates start of path) and the program stack (left). Black indicates main stack whereas blue indicates use of a learned library function. Non-library learning programs do not use learned library functions and thus function only on the base stack. (C) Human-generated language descriptions typically have lower description length than the synthetic descriptions. Analogously, programs with library learning are on average shorter than programs with just the base DSL. (D). Boards that elicit shorter more abstract descriptions from humans also elicit shorter programs, indicating a correspondence between the abstractions human use in language and those from with library learning.

gram's board and not in the target board.[1] The program is therefore strongly encouraged to mark all the tiles that are marked in the target, and strongly discouraged from marking additional tiles that are not in the target. Second, there are penalties that encourage the program to replicate the target board efficiently. There is a penalty of $-1$ for each instance of `move`, `left`, and `right` when the pen was down, encouraging programs to draw boards in few steps. The score of a program is set as the best score achieved by the program starting from any initial location $\max_l s(\rho|l)$.

In the first sleep phase, DreamCoder trains a neural network that is used to guide program search – this *recognition model* does so by taking the board as input and predicting programs that are likely to solve that board based on previous successful solutions. The recognition model consists of a 16-channel $3 \times 3$ convolutional layer, followed by two fully-connected layers with 64 units each, and ending with a linear layer. To generate a vector-based representation of the boards that contains the information necessary for program induction, we use the final hidden layer of the recognition model as the representation of the board. This embeds each board into a dense vector, similar to the natural language embedding of each board.

In the *library learning* sleep phase, DreamCoder grows the DSL by adding library functions based on reused components of successful programs found during wake (see Fig. 4B for an example). Specifically, it adds program components that minimize the description length of both the current program library and each successful program, once rewritten to use this new library component. These learned library abstractions serve to compress programs found in previous wake cycles and better enable the search for future programs in future wake cycles. We execute DreamCoder with and without library learning to test the usefulness of program representations with abstractions.

---

[1]This penalty is stronger because incorrectly marked locations can't be unmarked. "What's done cannot be undone." – Lady Macbeth

### 3.3 Comparing programs and language descriptions

We first consider the descriptions we obtained from humans. We found a wide range in the level of abstraction in human natural language descriptions. Some are more literal and verbose in their descriptions whereas others use more abstract concepts to compress their descriptions (Fig. 4A). On average, human-generated descriptions were shorter than synthetically-generated descriptions, which listed every red tile location (Fig. 4C). By analogy, library-learned concepts allow for compression of programs (Fig. 4B). Both human-generated language (vs synthetic) and library-learned programs (vs only primitives) have the same effect on enabling shorter description lengths (Fig. 4C).

We further found that the description lengths in human language and library-learned programs are significantly correlated (Fig. 4D) across boards. In other words, a board that is more compressible using human language abstractions was also more compressible with library learning over programs. This indicates that the abstractions used by humans in language overlap with the abstractions learned by Dreamcoder during library learning. This correlation is not significant when comparing human generated language description length with description lengths of programs without library learning (Fig. 4D). This indicates that the correlation may be due to the abstractions captured through library learning. An illustrative example can be seen in Fig. 1, where the program's used library function and the phrase "U shape" within in the human language description are actually the same concept. We verify this further by testing for the difference between these two correlations using a non-parametric permutation test. The correlation between human-generated language description length and library learning program description length is statistically significantly higher ($p = 0.0034$) than the correlation between human-generated description length and non-library learning program description length.

Not only do the representations between language and programs have correlated description lengths, but the actual representations learned can be similar as well. We consider a sample of three relatively simple boards in Fig. 5 and conduct a representational similarity analysis. We find that the similarity structure of these boards overlap across modalities – i.e they are similar between human-generated language and library-learned programs, as well as between synthetic language and primitive programs. Conversely, these matrices don't look very similar within modality. That is, the matrices for abstract human-generated language don't look very similar to the matrices for non-abstract synthetic language, and similarly for programs. This indicates that modality is not the key driver of similarity in representation space; rather, the level of abstraction can be more salient. This also indicates that the process of abstraction changes the representation in similar ways across language and programs.

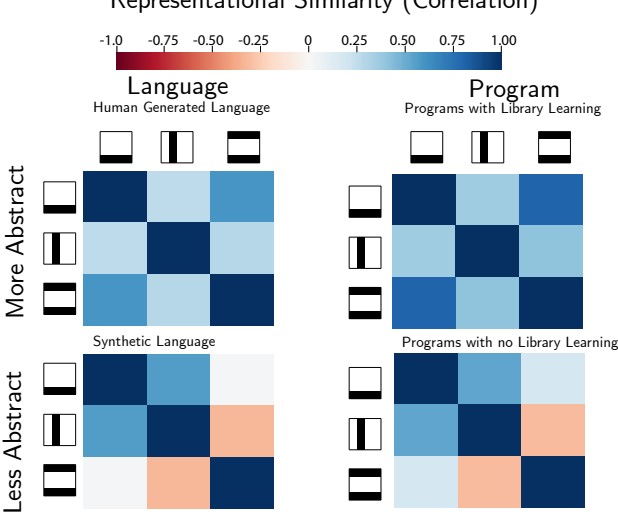

Figure 5: **Representational Similarity Between Language and Programs.** Similarity matrices (correlation matrices for respective embeddings) for a sample of human-generated boards with line motifs for human-generated language, programs with library learning, synthetic language, and programs without library learning. Abstract representations' similarity profiles (human language, library programs) across modalities are more similar than those of within-modality representations (human language+synthetic language, library learning programs+non-library learning programs).

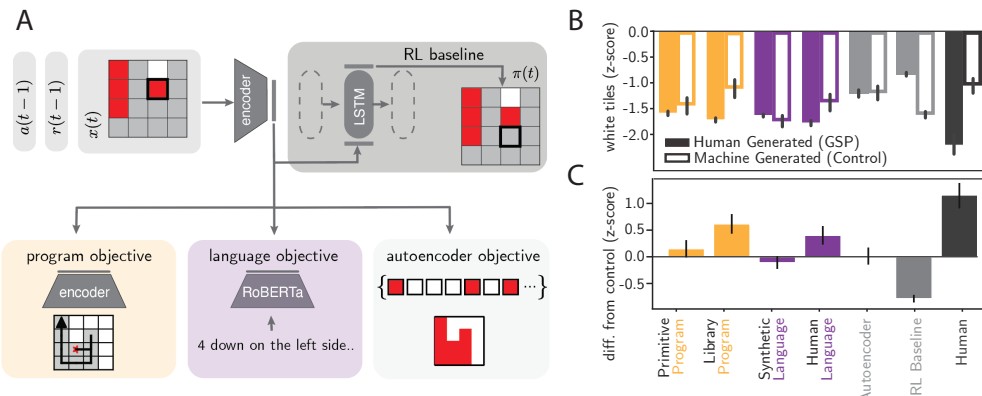

Figure 6: **Grounding Mechanism** (A) Mechanism for grounding agents using specific representations. We have the agent's encoder output simultaneously predict a task embedding while performing the task. These task embeddings can be from programs (embeddings from the recognition model of DreamCoder), language (embeddings from RoBERTa), or the flattened board itself (autoencoder baseline). (B). Performance on previously unseen GSP (colored bars) and control tasks (white bars with colored borders) from grounding agents with different task representations; lower is better (since performance is based on white tiles revealed). Error bars are 95% confidence intervals across different 15 trained agents (for humans, across 50 different participants). (C). Gap between performance on human-generated and machine-generated tasks. Higher gap indicates more human-like behavior of the model.

## 3.4 Grounding agents in language and program representations

In order to guide the agent to learn a human-like inductive bias, we introduce a *task grounding term* to the loss function: loss $= L^{PPO}(\theta) + c_{task}L^{task}(\hat{\psi}_\theta, \psi)$. Here $L^{PPO}(\theta)$ is the original PPO loss function, $\theta$ is the current model's parameters, $c_{task}$ is a hyperparameter coefficient that weights the task embedding loss $L^{task}$ (see Fig. 6A for the types of task embeddings we used), $\psi$ is the task embedding, and $\hat{\psi}_\theta$ is the agent's prediction of the task embedding. The task embedding loss $L^{task}$ is the Mean Squared Error (MSE) between the agent's predicted and actual task embedding. This forces the agent to simultaneously predict an embedding of the hidden task state (i.e. the underlying board) as well as come up with the best possible action for the current observation (see Fig. 6A).

Like the Baseline RL agent in Fig. 3A, we trained our learners on a training set from the human-generated boards and evaluated its performance on a held-out test set of human-generated and machine-generated boards. We trained different agents to predict different kinds of task embeddings during training (Fig. 6A) based on language (both synthetic and human-generated language) and programs (both no library learning and library learning). We refer to co-training on predicting these auxiliary representations as "grounding" the agent on the corresponding representation. The term "grounding" is often used in work that trains agents to associate data in one modality (i.e. text instructions) to another (i.e. visual images) [41–47]. We also co-trained on directly predicting a flattened version of the underlying board stimulus, which is equivalent to co-training the agent on an autoencoder loss.

We set out to test whether grounding meta-learning agents on these task embeddings will result in them producing more human-like performance. We know humans perform better on GSP boards than control boards (Fig. 3B), while the generic meta-RL agent does the opposite. Performing better on the GSP boards and worse on the control boards would therefore indicate more human-like behavior. To compare task performance across different task distributions or different models, we use a non-parametric bootstrap independent samples test to test for significance.

We see that **grounding with human-generated descriptions leads to a human-like inductive bias** (Fig. 6B). Training on this auxillary loss led to an increase in performance at the GSP boards relative to the baseline agent ($p < 0.0001$). The human-language grounded agent also performed significantly better at the GSP boards than the control boards ($p < 0.0001$), just like humans do. In

contrast, this auxiliary loss significantly *reduced* performance on the machine-generated control boards (as compared to the baseline agent, $p = 0.021$). This indicates that the auxiliary loss did not generically improve performance across the board, but rather improves performance selectively in the human-generated boards, and worsens it in the machine generated boards. In other words, it makes the agent *more human-like*. The human-language grounded agent also does the best at the human-generated tasks across all models.

The autoencoder agent is substantially better on the GSP boards than the original agent ($p < 0.001$). However, its performance does not differ significantly between the human and machine-generated tasks ($p = 0.252$), a performance pattern different from that of humans. Similarly, we also find that **grounding on synthetic descriptions does not lead to human-like bias** and instead has an effect similar to that of the autoencoder loss – it improves performance uniformly rather than selectively. In fact, the agent using synthetic text does better in the control distribution than the GSP distribution ($p = 0.0016$), which is qualitatively similar to the performance pattern of the original agent (Fig. 3B), and distinctly dissimilar from human behavior.

We now move to examining the agents co-trained with program representations. We found that **grounding on library learning programs leads to a human-like inductive bias** (Fig. 6B). Specifically, although grounding on non-library program task embeddings does numerically better on human-generated tasks than machine-generated tasks, this difference is not significant ($p = 0.11$). This difference, however, is highly significant when grounding on library-learning program task embeddings ($p < 0.0001$). In addition, grounding on library learning program task embeddings does significantly better at the GSP tasks than grounding on non-library learning program embeddings ($p < 0.0001$). Just as grounding on human language produces more human-like inductive bias than synthetic language, grounding on library learning programs produces more human-like inductive bias than grounding on programs without library learning.

This points to evidence that the **level of abstraction of the task embeddings influences the extent to which the agent acquires a human inductive bias**, as indicated by the differences in performance among low (synthetic for language, no library learning for programs) and high-level (human-generated language and library learning programs) agents. In human-generated descriptions, humans write about abstract concepts (e.g. lines, shapes, letters, etc). Similarly abstract concepts can be learned as library functions (see Fig. 4B). The process of abstraction in language and programs allows for compression of the respective description and induced program (Fig. see 4C). The embeddings of these representations that were compressed through abstract concepts may therefore be distilling useful concepts into our meta-learning agent that enables it to acquire a human-like inductive bias.

## 4 Discussion

In this work, we argue that *language descriptions and program abstractions can act as repositories for human inductive biases* that may be distilled into artificial neural networks. To test this idea, we considered a tile-revealing task with 2D grids that were directly sampled from human priors. We found that grounding on either human-generated language or programs with library learning not only *improves* machine performance on tasks that people perform *well* at, but also *impairs* performance on tasks that people perform more *poorly* at. Although the idea of co-training agents on language to shape their representations has been explored before, most works utilize synthetic descriptions [17, 21]. Our work suggests *human-generated language* can lead to more human-like performance than more literal synthetic language descriptions [16], because human-generated descriptions contain information about abstract concepts (e.g. lines, shapes, letters, etc) that compress description length and are reflected in tasks directly sampled from human priors, therefore better capturing human inductive biases. Although we used synthetic descriptions that are as literal/devoid of abstractions as possible to test this point, synthetic descriptions built to contain such abstractions could work as well. The challenge, then, in using such synthetic descriptions would be coming up with the correct abstractions to build in, rather than crowdsourcing such abstractions from human participants as we do in this work.

We also show evidence that co-training artificial agents on representations from program induction results in learning human inductive biases just as co-training on language does. Programs induced with library-learned concepts [24] are analogous to language in that adding library-learned con-

cepts to the initial DSL enables compression of description length just as how humans use abstract concepts to compress description length. Like human language, programs with library abstractions better enable learning of human-like inductive biases. Program induction can sometimes be difficult due to the combinatorial explosion of the program space [48] and the non-trivial decision of picking a DSL. Co-supervision of *tabula rasa* artificial agents using representations from program induction can be a useful way to build agents with both the flexibility of neural networks and the human-like priors that program induction can provide. Additionally, although we investigate language and programs as separate but analogous representations that can guide the training of artificial agents, recent work has also learned joint compositional generative models over program libraries and natural language descriptions [18]. An exciting direction for future work is to combine our objectives, utilizing joint program and language representations to instill human inductive biases in artificial agents.

In this work, we use a constrained task paradigm on 2D grids, which gives us the advantage of being able to directly study samples of human and machine priors on the task space (Fig. 2). Although some abstractions we found may be specific to this domain, the literature in cognitive science [4] provides good reason to expect that the effectiveness of our approach in this more artificial domain could translate to abstractions in real-world domains that are natural for humans. Some specific domains, such as mathematical reasoning, planning, or puzzle solving, may even more obviously necessitate abstractions [49]. It will be exciting future work to use our approach to empirically validate this conjecture by determining the space of tasks for which co-training with language and program abstractions improves an artificial neural agent's performance. This future work will be an exciting opportunity both for studying human intelligence by getting a better context for what the role of abstraction is in everyday cognition as well as improving machine intelligence by expanding the capabilities of current neural agents. However, there is still work to do to determine how language and programs can influence acquisition of human inductive biases in more scaled up, naturalistic, and real-world settings.

On the language side, the main bottleneck is that collecting human descriptions with sufficient coverage can become prohibitively expensive as the state space grows larger. One potential remedy would be to employ active learning methods to focus language elicitation on the most relevant abstractions in the environment and to explore data augmentation approaches to make these descriptions go further. For example, a vision-language model [50] could be fine-tuned on the human descriptions to generate sufficiently high-quality descriptions for new states. There is some new work showing that even a small amount of language can be used for models to perform visual grounding on large environments [51]. This approach to scaling language may require scoring descriptions for their quality in describing the relevant abstractions in the environment, which is less time-consuming for human participants than producing such descriptions.

On the program side, the need to define *ad hoc* base DSLs for each domain is an important limitation of any approach relying on program abstractions. An important direction for this area is to develop a 'canonical' basis for program representations that may apply across many domains. For example, a graphics engine may provide one such general-purpose DSL to synthesize state abstractions for larger and more realistic visual environments. This kind of base DSL could be seen as an extension to the generic 2D vector graphics used by Ellis et al. 52, to induce graphics programs from hand-drawn images. A potential bottleneck here is that library learning can be especially computationally expensive for larger environments, but several algorithmic improvements have already emerged that make this direction more promising [53].

Strong inductive biases towards abstract structure allow humans to generalize to novel environments without much experience [4]. By instilling these biases in our artificial agents, we can work towards enabling machines to demonstrate human-like general intelligence.

## Acknowledgements

S.K. was supported by NIH T32MH065214 during the duration of this work. R.D.H is supported by a C.V Starr Fellowship. This work was supported by the the U.S. Army Research Office grant W911NF-16-1-0474, the DARPA L2M Program, ONR grant N00014-18-1-2873, and the John Templeton Foundation. The opinions expressed in this publication are those of the authors and do not necessarily reflect the views of the funders.

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
