# OpenReview forum: "Using natural language and program abstractions to instill human inductive biases in machines"
_NeurIPS.cc/2022/Conference — NeurIPS 2022 Accept_

### Official Review · Reviewer_zreP · 2022-07-06

**Rating:** 8
**Confidence:** 3
**Soundness:** 3 good
**Presentation:** 4 excellent
**Contribution:** 3 good

**Summary:**

This paper presents an experimental setup and set of results that analyze (1) whether RL agents exhibit similar behavior to people in guessing which squares on a board are colored, in two different distributions of boards: human-elicited, and machine-produced, and (2) whether more human-like behavior (good performance on human-elicited; poor on machine-produced) is obtained by the RL agent when co-training the it using an auxiliary loss that predicts one of 4 types of representation: human-written board descriptions, low-level synthetic language board descriptions, program abstractions induced by the DreamCoder algorithm, or program primitives (which span 2 modalities and 2 rough levels of abstraction). The paper's main finding is that regularizing RL agent training with either human-generated language or induced program abstractions make the agent's behavior correspond more closely to human behavior.

**Questions:**

Q1) 123- I was pretty unclear on how meta-learning is used in the agent training. Is a meta-learning algorithm used (it doesn't seem to be; PPO seems to be applied in each individual task), or is it a meta-learning architecture? If the second, does the LSTM model the temporal nature of individual tasks (recurrence is across timesteps for individual agent actions) or the temporal nature of learning across tasks?

Q2) What are 120 LSTM units? Does this mean an LSTM with hidden dimension 120, or 120 different LSTM cells, or 120 unrolled time steps (or something else)?

*Suggestions*

The paper shows similar effects from natural language abstractions & learned program abstractions, but it would be interesting to also investigate whether there is a direct correspondence between these modalities, e.g. using alignment methods similar to those in Andreas et al. 2017, Translating Neuralese or Andreas 2019, Measuring compositionality in representation learning [I feel I should disclaim here that I am not an author of these works :) ].

Claims in the results descriptions / discussion that I'd suggest revising/rewording:
- 275: "improves performance uniformly": if I'm reading the results correctly, autoencoder seems to decrease the performance of the RL baseline on the machine-generated boards
- 302-303: I may have misunderstood, but the library learning isn't grounded in language in the current work, is it (i.e. the paper doesn't do a LAPS-like, Wong et al. approach) -- it's just the RL agent that is grounded?
- Mu et al. does use human-generated language for the Birds dataset. This would also be a good place to cite Andreas et al., Learning with Latent Language
- 312: this is describing the results in Fig 5B, right? there are really only 2 bits in the comparison (relative to the RL agent with no representation prediction, GSP goes up; control goes down), and so this wasn't totally convincing to me


*Minor suggestions*
- line 51: is *a* meta-learning
- 59: define the auxiliary representation here
- 93-95: this seems intuitive, but it could help to give some stats to make it concrete -- what does it mean to respect most of the stat properties?
- 240: it wasn't clear until seeing Fig 6 that "task attribute" can include representations of the language or program.
- 250-252: I confess that having only these citations bugged me a bit, given the long tradition of work on language grounding before 2018.
- Fig 5 caption: "across modalities" was a bit confusing to me: it's the correlation matrices that are being compared across modalities, right, not the individual representations? (i.e. language embeddings are never compared to program embeddings)

**Limitations:**

The discussion of the limitations at the end of the paper was overall thorough and clear. The main things I would suggest addressing are in the weaknesses section above.

One additional point of uncertainty for me was what typifies the "machine prior". It's not at all clear to me that an agent should do better on the control distribution than on the human distribution, particularly given what seem to be differences in the architecture used to produce the distribution and the one used in the RL agent. It would strengthen the paper to explore this more. The nearest neighbor heuristic also encodes a human-like prior in some sense, and since many results are reported relative to this heuristic, I wonder if this could be stacking the cards against the agent.


**Strengths And Weaknesses:**

*Strengths*

S1) I found the paper thought-provoking -- it's likely to stimulate future work on program induction and grounding language to abstraction. I'm likely to recommend this paper to others working in these areas, as well as to point to it as a nice example of a paper that combines cognitive science investigation with solid ML techniques.

S2) The experiments and analysis were overall very well-designed and carefully controlled (but see below for some minor suggestions) and thorough, anticipating and heading off several concerns that I had.

S3) I thought the choice of setting was a strength -- the grid world environment seems rich enough to produce sophisticated natural language and programs at varying levels of abstraction, while still being simple enough to afford controlled experiments, in particular collecting the GSP boards.

S4) The paper was extremely well-written and clear. I enjoyed reading it.


*Weaknesses*

W1) The use of language in the paper is not quite as well-developed as the use of program abstractions. There's a stark contrast between the nature of the synthetic language and human-written language -- not just in the level of abstraction, but also in being human-written versus synthetic. The synthetic language is likely far out-of-distribution for the RoBERTa model, and the language representations (if I understood correctly) weren't induced for any of the tasks in this paper (the RoBERTa model was frozen). I would have appreciated a finer-grained analysis that compared different levels of abstraction in the human-written language. But I don't think this is a crucial weakness.

*Edit after response:* In the response, the authors authors point out differential effects of synthetic language on human vs model boards, which I agree give evidence that improvements from human language aren't just due to being OOD for RoBERTa. So I'm even less concerned about this now.

W2) I worry a bit that the findings here are bound to depend on the program abstractions and the language elicited. I don't think this is a crucial weakness because the paper was pretty careful here (primitives and synthetic language, while similar in design, are natural for this domain; and the language is human-elicited and program abstractions are learned from just a few low-level primitives) -- but I'd be more convinced if the main findings were also shown to hold in another domain. But again I don't think this is a crucial weakness, there's enough here that that could be followup work.

*Edit after response:* I agree with the authors' points that these would be exciting for future work but are out-of-scope for the submission.

I've updated my score to an 8 (from a 7).

---

> ### Author Response · Authors · 2022-08-02
> **Response to Reviewer zReP Part 1**
>
> We thank the reviewer for their positive comments and all their constructive feedback. Below is a response to the questions/concerns  raised.
>
> > W1) The use of language in the paper is not quite as well-developed as the use of program abstractions. There's a stark contrast between the nature of the synthetic language and human-written language -- not just in the level of abstraction, but also in being human-written versus synthetic. The synthetic language is likely far out-of-distribution for the RoBERTa model, and the language representations (if I understood correctly) weren't induced for any of the tasks in this paper (the RoBERTa model was frozen).
>
> We agree the synthetic language may be more out-of-distribution for the RoBERTa model than the human-written language, and incorporating finetuning is an important next step. For the purpose of our experiments, however, the synthetic distributions being OOD doesn’t fully account for the effects we found. In particular, we  note that the performance boost for human-written language is selective in that it only helps for human-generated boards, while for machine-generated boards, the synthetic language performs better than the human language. This suggests that the synthetic language still provides a useful signal for the model. We would not necessarily expect such selectivity in performance boost if the only difference between synthetic and human language was that one is more in distribution for the large language model.
>
> > I would have appreciated a finer-grained analysis that compared different levels of abstraction in the human-written language. But I don't think this is a crucial weakness.
>
> This is a great point. In future work we would like to more systematically dig into the different levels of abstraction in the human-written language, ideally using a human experiment where we rate descriptions based on description length vs reconstructability (i.e. can someone reconstruct the board given the description?). This experiment would provide a more convincing way of quantifying the level of abstraction (and its usefulness) in the descriptions than simply being elicited by different prompts. It would then be interesting to show what level of abstraction helps the agent learn human inductive biases the most (i.e. too little and too much abstraction both may not be so helpful).
>
> > W2) I worry a bit that the findings here are bound to depend on the program abstractions and the language elicited. I don't think this is a crucial weakness because the paper was pretty careful here (primitives and synthetic language, while similar in design, are natural for this domain; and the language is human-elicited and program abstractions are learned from just a few low-level primitives) -- but I'd be more convinced if the main findings were also shown to hold in another domain. But again I don't think this is a crucial weakness, there's enough here that that could be followup work.
>
> We agree that our task is tailored to visualizing abstract human inductive biases. Now that we’ve validated our approach using a careful experimental setup in which such abstractions are easily visualized, we can set out to scale up this approach in future work on larger, more real world and natural domains. For larger environments, this may involve automating the language description generation with the use of vision-language models or using graphics programs DSL’s to synthesize photorealistic images with structure that can apply across many domains (see our response to the first point for a larger discussion on this). Additionally, human abstractions in some domains will help more than others. Some domains, such as mathematical reasoning (as the reviewer brought up), planning, or puzzle solving even more obviously necessitate abstractions. From the literature in cognitive science (e.g. Lake et al. 2017), we believe these domains correlate with tasks that humans are good at and typical neural network agents are not so good at. It will be exciting future work to use our approach to test this correlation by evaluating which other tasks co-training with language and program abstractions can improve neural agent’s ability. Doing so will be an exciting opportunity both for studying human intelligence by getting a better context for what the role of abstraction is in everyday cognition as well as improving machine intelligence by expanding the capabilities of current neural agents.
>
> Works cited:
>
> * Lake, B. M., Ullman, T. D., Tenenbaum, J. B., & Gershman, S. J. (2017). Building machines that learn and think like people. Behavioral and brain sciences, 40.

---

> ### Author Response · Authors · 2022-08-02
> **Response to Reviewer zReP Part 2**
>
> > Q1) 123- I was pretty unclear on how meta-learning is used in the agent training. Is a meta-learning algorithm used (it doesn't seem to be; PPO seems to be applied in each individual task), or is it a meta-learning architecture? If the second, does the LSTM model the temporal nature of individual tasks (recurrence is across timesteps for individual agent actions) or the temporal nature of learning across tasks?
>
> We apologize for the paper not clearly stating the exact connection with meta-learning. Indeed, we are using a meta-learning architecture, and the LSTM is modeling the temporal nature of individual tasks. In the supplement , we have included this explanation as well as relevant references that details this connection in the revised paper (reproduced below):
>
> “Meta-learning can be considered as a bi-level optimization problem where there is an outer-loop of learning in which the model learns a useful inductive bias across different tasks of the same task distribution and an inner-loop of learning which takes that inductive bias and rapidly learns or adapts within a specific task (Hospedales et al. 2020). In recurrent-based meta-reinforcement learning architectures like ours, in which tasks are fed sequentially to the model, the outer loop is explicitly implemented as a reinforcement learning algorithm that updates the weights across tasks and the inner loop is implicitly implemented as a separate reinforcement learning algorithm in the activation dynamics of the recurrent network (Wang et al. 2016; Botvinick et al. 2019) that employs fast adaptation within a specific task. See (Ortega et al. 2019) for a formal explanation of this adaptation. In our case, the LSTM weights are updated in the outer loop across different grids to give the LSTM a useful prior learned from patterns or abstractions seen across different grid tasks. The activation dynamics within the LSTM utilizes this prior, along with the history of observations within the episode, to implement a fast algorithm to solve the current grid task for the inner loop.  See (Ortega et al. 2019) for a formal explanation of this adaptation.”
>
> > Q2) What are 120 LSTM units? Does this mean an LSTM with hidden dimension 120, or 120 different LSTM cells, or 120 unrolled time steps (or something else)?
>
> We apologize for being vague here. We meant that we used an LSTM with 120 different *cells.* We have clarified this point in the revised document.
>
> > The paper shows similar effects from natural language abstractions & learned program abstractions, but it would be interesting to also investigate whether there is a direct correspondence between these modalities, e.g. using alignment methods similar to those in Andreas et al. 2017, Translating Neuralese or Andreas 2019, Measuring compositionality in representation learning [I feel I should disclaim here that I am not an author of these works :) ].
>
> We agree that this is an important future direction that we would be very interested in pursuing. In addition to the works mentioned by the reviewer, Wong et al. 2021 (ICML) uses a similar joint language and program idea as an objective and Wong et al., 2022 (CogSci) uses it as an evaluation metric.
>
> > One additional point of uncertainty for me was what typifies the "machine prior". It's not at all clear to me that an agent should do better on the control distribution than on the human distribution, particularly given what seem to be differences in the architecture used to produce the distribution and the one used in the RL agent. It would strengthen the paper to explore this more.
>
> We agree that it is important to investigate what typifies the machine prior. We believe a systematic exploration of different combinations of architectures that produce the distribution and use the distribution during training would allow us to answer this question. We are excited to explore this in future work.
>
> Works cited:
>
> * Wong, C., Ellis, K. M., Tenenbaum, J., & Andreas, J. (2021, July). Leveraging language to learn program abstractions and search heuristics. In International Conference on Machine Learning (pp. 11193-11204). PMLR.
>
> * Wong, C., McCarthy, W. P., Grand, G., Friedman, Y., Tenenbaum, J. B., Andreas, J., ... & Fan, J. E. (2022). Identifying concept libraries from language about object structure. arXiv preprint arXiv:2205.05666.

---

> ### Author Response · Authors · 2022-08-02
> **Response to Reviewer zReP Part 3**
>
> > The nearest neighbor heuristic also encodes a human-like prior in some sense, and since many results are reported relative to this heuristic, I wonder if this could be stacking the cards against the agent.
>
> We agree with the reviewer that the nearest neighbor heuristic has a human-like prior (spatial proximity). However, it is not necessarily the primary human-like prior we are interested in studying (abstraction). This z-score metric was introduced in Kumar et al. 2021 ICLR to control for this proximity prior so that the prior of interest can be studied and we follow suit in this work.
>
> We think it’s less likely that this could be stacking the cards against the agent, because both humans and all agents are evaluated on all task distributions under the same metric for comparison. We also believe that, since the agents all use a conv layer, it would naturally contain this bias of spatial proximity (Kumar et al. 2021 ICLR has a discussion on these points in the section titled “Bias towards spatial proximity.”).
>
> > 275: "improves performance uniformly": if I'm reading the results correctly, autoencoder seems to decrease the performance of the RL baseline on the machine-generated boards
>
> Yes, we thank the reviewer for catching this error and we have amended the claim accordingly in the new submission.
>
> > 302-303: I may have misunderstood, but the library learning isn't grounded in language in the current work, is it (i.e. the paper doesn't do a LAPS-like, Wong et al. approach) -- it's just the RL agent that is grounded?
>
> Yes, the reviewer is right. We used program induction (including library learning) as a separate training signal for the RL agent (independent of also using language as a training signal for the agent). We apologize that this was ambiguous and have made this more explicit in the new submission.
>
> > Mu et al. does use human-generated language for the Birds dataset. This would also be a good place to cite Andreas et al., Learning with Latent Language
>
> We thank the reviewer for catching this and have made the appropriate citations accordingly in the new submission.
>
> > 312: this is describing the results in Fig 5B, right? there are really only 2 bits in the comparison (relative to the RL agent with no representation prediction, GSP goes up; control goes down), and so this wasn't totally convincing to me
> We have softened this claim in the new submission.
>
> > line 51: is a meta-learning
>
> > 59: define the auxiliary representation here
>
> > 93-95: this seems intuitive, but it could help to give some stats to make it concrete -- what does it mean to respect most of the stat properties?
>
> > 240: it wasn't clear until seeing Fig 6 that "task attribute" can include representations of the language or program.
>
> > 250-252: I confess that having only these citations bugged me a bit, given the long tradition of work on language grounding before 2018.
>
> We appreciate all these suggestions. They have been incorporated into the revised document.
>
> > Fig 5 caption: "across modalities" was a bit confusing to me: it's the correlation matrices that are being compared across modalities, right, not the individual representations? (i.e. language embeddings are never compared to program embeddings)
>
> Yes, this is correct in that we are comparing the correlation matrices and not the specific representations. We have made this clearer in the revised paper.
>
> Works cited:
>
> * Kumar, S., Dasgupta, I., Cohen, J., Daw, N., & Griffiths, T. (2021). Meta-Learning of Structured Task Distributions in Humans and Machines. In International Conference on Learning Representations.

---

### Official Review · Reviewer_QkpB · 2022-07-07

**Rating:** 8
**Confidence:** 4
**Soundness:** 3 good
**Presentation:** 3 good
**Contribution:** 3 good

**Summary:**

The aim of this paper is twofold: (1) showing humans and machines use different inductive biases, and (2) showing that machines can learn to generalise more like humans do by distilling human inductive biases into them during training.

As a proxy for human-like inductive bias, the authors use the setup from [11] with two distributions of tile revealing tasks: one reflecting human inductive biases and one so-called "metamer" task, that is similar to the human-biased task in statistics, but is generated differently. The task reflecting human inductive biases (called GSP boards) are 2D grids with shapes sampled directly from humans. The metamer task uses the exact same sampling procedure, but instead of a human in the loop, there is a machine in the loop (called machine-generated boards). The task the agent is evaluated on is revealing tiles on a 2D grid to uncover a certain shape of red tiles, while revealing as little white tiles as possible. For the human-like tasks, imagine U-like, I-like, or pyramid-like shapes on a board, and the machine-generated boards are similar except that it's not symmetric and smooth shapes. Humans perform better on the GSP boards than on the machine-generated boards.

Towards aim (1), showing humans and machines use different inductive biases, the authors train an RL agent with PPO on a set of GSP boards (i.e. red shapes on a 2D grid), and test on a held-out set of GSP boards as well as on machine-generated boards. Even though the agent has been trained on GSP boards, the agent performs better on the machine-generated boards than on the GSP boards. The authors hypothesise that this is because humans and machines use different biases to solve the task, and even though the machine is trained on the human-generated boards reflecting the human inductive biases, because the machine-generated boards have similar statistics to the GSP boards, they are better at those

Towards aim (2) the authors train the baseline model again but with an auxiliary term added to the loss. The encoder of the baseline model is, besides learning the policy, trained to reconstruct a representation of the grid. Four different representations are experimented with: (1) synthetic language describing the board (red tile on row x1 col y1, red tile on row x2 col y2, etc.), (2) crowdsourced natural language describing the board (u-like shape in the bottom left, etc.), (3) a program constructed of a bunch of primitives from a DSL learned with DreamCoder (primitives like move, pen-up, pen-down, fork, left, right, so programs end up following the shape), (4) a program  learned with DreamCoder + library learning (meaning certain re-usable learned programs are added to the DSL). The authors show that the descriptions lengths of natural language descriptions and programs with library learning are correlated, and that some abstractions are even similar (i.e. humans say U-shape, and DreamCoder adds a U-shaped-program to the DSL). Additionally, they train the encoder to simply reconstruct the flattened grid, called the autoencoder objective.
Results show that for the autoencoder objective, synthetic language objective, and primitive program objective (DreamCoder without library learning) don't distill human-like inductive biases, but natural language descriptions and programs with re-used components do. This indicates that the level of abstraction of the task embeddings is important for distilling human inductive bias. Human-generated language is important and works better than synthetically generated language.

**Questions:**

- I don't see under what definition this method uses meta-learning. As far as I know [2], meta-learning always has some meta-phase (i.e. doing some learning over the task distribution) to learn a meta representation (e.g. initialisation of parameters like in MAML) and an inner phase on a task sampled from the task distribution. This method does have a task distribution, but there's no notion of meta-learning as far as I can see. I could have an outdated idea of what constitutes meta-learning nowadays, so adding it to the questions :)

- Nit 1: Can you add how many runs you did to get the performance in variance, and add in the bar chart figures that the bars are variance over different runs / different humans?

- Nit 2: I can't parse line 249-250

- Nit 3: I personally found library learning in the abstract slightly confusing because I didn't know the term. Generally, I think the abstract could use some work on clarity.

[2] Meta-Learning in Neural Networks: A Survey; Timothy Hospedales, Antreas Antoniou, Paul Micaelli, Amos Storkey

**Limitations:**

The two main limitations are properly addressed by the authors in the discussion, but I would like to see some more speculation on the direction of the future work and the scalability of this method. Party copied from the strengths&weaknesses part above my suggestions are:

- Would like to see more discussion on scaling this approach to real-world tasks. There are many tasks for which humans are thought to use program-like representations (e.g. mathematics, counting, addition, etc., see [1] below).

- Perhaps vision-language models could be used to generate the language descriptions to get around the need for crowdsourcing them.

- I wonder whether these results transfer to problems where the solution is less obviously constructed from the abstractions, perhaps you could discuss this

**Strengths And Weaknesses:**

Strengths:
- This paper is very carefully and clearly written, with clear figures supplementing understanding.
- The experimental setup is sound and convinces the reader of the main points.
- The result that reproduces Kumar et al. in showing that humans have different inductive biases is an interesting and (to me) surprising result, and even though it's not a novel result it's still important in RL to replicate results from other papers with different algorithms (PPO vs A2C). Different training algorithms can give very different results in RL, and the fact that this result is stable under PPO and A2C strengthens it.
- Interesting main findings; natural language descriptions and programs learned with DreamCoder + library learning are correlated in certain features, and even share abstractions, and these representations are also the ones that make the model generalise more like humans do.
- I'm very excited about the potential of using the abstractions that are naturally present in human language for better generalisation, this is a very important research direction and I would like to see more in this area, this paper is a great step in that direction

Weaknesses:
- It is mentioned in the discussion, but the weakness of this method is that either a DSL needs to be created or natural language descriptions of each task in the dataset. The former isn't as expensive but might be difficult for more realistic tasks, and the latter is expensive. It would be nice to see some more discussion on the feasibility of scaling this approach to real-world tasks. There are many tasks for which humans are thought to use program-like representations (e.g. mathematics, counting, addition, etc., see [1] below). Also, perhaps vision-language models could be used to generate the language descriptions to get around the need for crowdsourcing them.
- Related; the tile-revealing task is a task very much tailored to literally visualising human-like inductive biases. Basically, in this task the solution is a sort of sum of the abstractions (e.g. U-like shape plus some red tile somewhere). Would these results transfer to problems where the solution is less obviously constructed from the abstractions?
- This is a very minor point, and maybe more of an opinion than a weakness, but in the discussion you claim that your work suggests "not all language is created equal: human-generated language leads to more human-like performance than synthetic language descriptions", but this seems like more of an artefact of the particular type of synthetic language you chose, namely one without any abstractions, simply listing the tiles. Probably a synthetically generated description that would take into account shapes like lines (which are easily automatically detected) would work as well as natural language. In any case, I do agree that natural language is more suited to this, I just think the comparison shouldn't be synthetic versus natural but rather learned/natural abstractions (language/library learning) versus hardcoded synthetic abstractions (that will cover less of the abstractions in the data).

[1] The Child as Hacker, Joshua S. Rule, Joshua B. Tenenbaum, and Steven T. Piantadosi

---

> ### Author Response · Authors · 2022-08-02
> **Response to Reviewer QkpB Part 1**
>
> We thank the reviewer for their positive comments and constructive feedback. Below is a response to the questions/concerns raised.
>
> > It is mentioned in the discussion, but the weakness of this method is that either a DSL needs to be created or natural language descriptions of each task in the dataset. The former isn't as expensive but might be difficult for more realistic tasks, and the latter is expensive. It would be nice to see some more discussion on the feasibility of scaling this approach to real-world tasks. There are many tasks for which humans are thought to use program-like representations (e.g. mathematics, counting, addition, etc., see [1] below). Also, perhaps vision-language models could be used to generate the language descriptions to get around the need for crowdsourcing them.
>
> We agree with the reviewer that this remains the most important next step for our work. We are excited to brainstorm potential steps to make the path to scaling more clear. The primary bottlenecks of our approach are the need to (1) collect language descriptions and/or (2) run program induction on the environment’s states in order to include them as co-training signals. We have added discussion of these avenues to our limitations paragraph in the discussion section.
>
> On the language side, we agree that a potential remedy would be moving to a weakly- or semi-supervised settings, finding the most relevant or ‘hardest’ abstractions in the environment, collecting a small number of language descriptions, and as the reviewer astutely suggests, fine-tuning vision-language models to augment these descriptions. Indeed, recent work (Yan et al. 2022 arXiv) has suggested that such models can be trained to do visual grounding with a small amount of captioned data and mostly uncaptioned data, for example. This approach may also necessitate the need for humans to provide independent ratings of description quality, which would be less expensive than producing de novo descriptions.
>
> On the program side, we agree that the need to define ad hoc base DSLs for each domain is an important limitation of any approach relying on program abstractions. An important direction for this area is to develop a ‘canonical’ basis for program representations that may apply across many domains. For example, a graphics engine may provide one such general-purpose DSL to synthesize state abstractions for larger and more realistic visual environments. This kind of base DSL may be seen as an extension to the generic 2D vector graphics used by Ellis et al., 2018, to induce graphics programs from hand-drawn images.
>
> > Related; the tile-revealing task is a task very much tailored to literally visualising human-like inductive biases. Basically, in this task the solution is a sort of sum of the abstractions (e.g. U-like shape plus some red tile somewhere). Would these results transfer to problems where the solution is less obviously constructed from the abstractions?
>
> We agree that our task is tailored to visualizing abstract human inductive biases. Indeed, our work intentionally set out to explore a potential solution to instilling inductive biases in a sound experimental setup where abstractions can be easily visualized and interpreted. Although some abstractions we found may be specific to this domain, the literature in cognitive science (e.g. Lake et al. 2017) provides good reason to expect that the effectiveness of our approach in this more artificial domain could translate to  abstractions in real-world domains that are natural for humans, which will be even more well-suited for language descriptions. Some such domains, such as mathematical reasoning (as the reviewer brought up), planning, or puzzle solving even more obviously necessitate abstractions. It will be exciting future work to use our approach to empirically validate this conjecture by evaluating which other tasks co-training with language and program abstractions improves neural agent’s ability.
>
> Works cited:
>
> * Yan, Chen, Federico Carnevale, Petko Georgiev, Adam Santoro, Aurelia Guy, Alistair Muldal, Chia-Chun Hung, Josh Abramson, Timothy Lillicrap, and Gregory Wayne. "Intra-agent speech permits zero-shot task acquisition." arXiv preprint arXiv:2206.03139 (2022).
>
> * Ellis, K., Ritchie, D., Solar-Lezama, A., & Tenenbaum, J. (2018). Learning to infer graphics programs from hand-drawn images. Advances in neural information processing systems, 31.
>
> * Lake, B. M., Ullman, T. D., Tenenbaum, J. B., & Gershman, S. J. (2017). Building machines that learn and think like people. Behavioral and brain sciences, 40.

---

> > ### Comment · Reviewer_QkpB · 2022-08-05
> > **Thanks for the detailed response**
> >
> > Thanks a lot for the detailed responses, all my points are adequately addressed. I agree with the authors that the weaknesses I mentioned in my initial review are better seen as avenues for future work. Very interesting to learn about this meta-learning approach for RL, I will read more about that. I will keep my rating at strong accept,  I'm excited to see this paper accepted and keep an eye on future work.

---

> ### Author Response · Authors · 2022-08-02
> **Response to Reviewer QkpB Part 2**
>
> > This is a very minor point, and maybe more of an opinion than a weakness, but in the discussion you claim that your work suggests "not all language is created equal: human-generated language leads to more human-like performance than synthetic language descriptions", but this seems like more of an artefact of the particular type of synthetic language you chose, namely one without any abstractions, simply listing the tiles. Probably a synthetically generated description that would take into account shapes like lines (which are easily automatically detected) would work as well as natural language. In any case, I do agree that natural language is more suited to this, I just think the comparison shouldn't be synthetic versus natural but rather learned/natural abstractions (language/library learning) versus hardcoded synthetic abstractions (that will cover less of the abstractions in the data).
>
> This is a great point and we very much agree. The synthetic descriptions we used are devoid of any abstractions (we made them as literal descriptions as possible) and it is very much plausible that synthetically generated descriptions with embedded abstractions (such as lines, shapes, etc) would better instill human inductive biases. We refrained from creating synthetic descriptions with specific abstractions ourselves since it would be an arbitrary choice, but we note that human descriptions do indeed contain these abstractions, and Dreamcoder libraries also pick up on these abstractions. We have thus reworded this observation in the discussion to focus on the ‘abstract language vs. literal language’ axis rather than the ‘natural vs. synthetic’ axis.
>
> > I don't see under what definition this method uses meta-learning. As far as I know [2], meta-learning always has some meta-phase (i.e. doing some learning over the task distribution) to learn a meta representation (e.g. initialisation of parameters like in MAML) and an inner phase on a task sampled from the task distribution. This method does have a task distribution, but there's no notion of meta-learning as far as I can see. I could have an outdated idea of what constitutes meta-learning nowadays, so adding it to the questions :)
>
> We apologize for the paper not clearly stating the exact connection with meta-learning. In the supplement, we have included this explanation as well as relevant references that details the connection in the revised paper (reproduced below):
>
> “Meta-learning can be considered as a bi-level optimization problem where there is an outer-loop of learning in which the model learns a useful inductive bias across different tasks of the same task distribution and an inner-loop of learning which takes that inductive bias and rapidly learns or adapts within a specific task (Hospedales et al. 2020). In recurrent-based meta-reinforcement learning architectures like ours, in which tasks are fed sequentially to the model, the outer loop is explicitly implemented as a reinforcement learning algorithm that updates the weights across tasks and the inner loop is implicitly implemented as a separate reinforcement learning algorithm in the activation dynamics of the recurrent network (Wang et al. 2016; Botvinick et al. 2019) that employs fast adaptation within a specific task. See (Ortega et al. 2019) for a formal explanation of this adaptation. In our case, the LSTM weights are updated in the outer loop across different grids to give the LSTM a useful prior learned from patterns or abstractions seen across different grid tasks. The activation dynamics within the LSTM utilizes this prior, along with the history of observations within the episode, to implement a fast algorithm to solve the current grid task for the inner loop. ”
>
> > Nit 1: Can you add how many runs you did to get the performance in variance, and add in the bar chart figures that the bars are variance over different runs / different humans?
>
> > Nit 2: I can't parse line 249-250
> We thank the reviewer for pointing these points of ambiguity out. We have clarified them more in the revised submission.
>
> > Nit 3: I personally found library learning in the abstract slightly confusing because I didn't know the term. Generally, I think the abstract could use some work on clarity.
>
> We apologize for the more confusing nature of the abstract and have refined it to be more clear.

---

### Official Review · Reviewer_czvj · 2022-07-13

**Rating:** 8
**Confidence:** 5
**Soundness:** 4 excellent
**Presentation:** 4 excellent
**Contribution:** 4 excellent

**Summary:**

This paper explores the intersection of abstraction, induction, language, and behavior. The goal of the paper is to instill human-like inductive biases into neural networks, in an attempt to improve generalization and performance. This is accomplished by using auxiliary tasks in an RL framework that attempt to elicit structured internal representations that match human representations, as measured by language-based representations of tasks.  The paper presents careful experimentation that involves a variety of tests of human and synthetic descriptions, and show that asking the agent to match human language generally induces more human like inductive biases.

**Questions:**

- Is it possible that the fact that DreamCoder has a DNN inside of it presents a confound that might result in its libraries and procedures being too similar to what an agent would learn directly anyway?

**Limitations:**

This paper suitably address both limitations and potential ethical concerns (which are minor).

**Strengths And Weaknesses:**

Wow. I really, really enjoyed this paper. It was thoughtful, extraordinarily well-written, and intellectually stimulating.

+ The idea of inducing human-like abstractions in general DNNs is important and potentially impactful

+ The paper is one of the most clearly written papers I've read in a long time

+ The experimental setup is complex, but clean, with each step naturally building on previous steps

+ The paper is intellectually enlarging, proposed new methodologies and tasks to make progress against a hard problem

- I wish there was some sort of clearer path to "scaling this up" to the next step -- a more complex agent, a larger DNN, a richer task.

- Ultimately, the paper only provides an /indirect/, behavioral measure of whether or not the internal representations of the DNN are human-like. It would be intriguing to see if any sort of /direct/ measure is possible (maybe by analyzing weights directly?).

---

> ### Author Response · Authors · 2022-08-02
> **Response to Reviewer czvj Part 1**
>
> We thank the reviewer for their positive comments and constructive feedback. Below we respond in detail to specific questions and concerns that were raised.
>
> > I wish there was some sort of clearer path to "scaling this up" to the next step -- a more complex agent, a larger DNN, a richer task.
>
> We agree with the reviewer that scaling to richer tasks remains the most important next step for our work.  Below we include some ideas about the feasibility of scaling. The primary bottlenecks of our approach are the need to (1) collect language descriptions and/or (2) run program induction on the environment’s states in order to include them as co-training signals. We focus on decision-making environments with visual observations, as this is the setting for our work.
>
> On the language side, the main bottleneck is that collecting human descriptions with sufficient coverage can become prohibitively expensive as the state space grows larger. One potential remedy would be to employ active learning methods to focus language elicitation on the most relevant abstractions in the environment and to explore data augmentation approaches to make these descriptions go further. For example, a vision-language model could be fine-tuned on the human descriptions to generate sufficiently high-quality descriptions for new states. There is some new work showing that even a small amount of language can be used for models to perform visual grounding on large environments (Yan et al. 2022 arXiv). This approach to scaling language may require scoring descriptions for their ‘quality’, which we could also conceivably accomplish using in another human experiment (scoring descriptions is less time-consuming for human participants than producing descriptions).
>
> On the program side, the main bottleneck is that library learning is not yet computationally tractable for larger environments. However, since submitting our work, several algorithmic improvements have already emerged that make this direction appear promising (e.g. Bowers et al., 2022, which introduces a top-down synthesis approach yielding 2-3 orders of magnitude improvements in performance).
>
> We have included a brief discussion of these directions in our limitations paragraph in the discussion section.
>
> > Ultimately, the paper only provides an /indirect/, behavioral measure of whether or not the internal representations of the DNN are human-like. It would be intriguing to see if any sort of /direct/ measure is possible (maybe by analyzing weights directly?).
>
> We agree that it would be interesting to more directly assess the similarity of internal representations of the agent with the internal representations of humans. The first step toward comparing internal representations is a rigorous evaluation of *external* behavior. But, as the reviewer hints, human-like external behavior is a necessary but not sufficient condition for human-like internal representations. Claims on alignment of internal representations are challenging to make using only choice observations. \
>
> The “gold standard” for measuring alignment between the agent’s internal representations and those of humans uses neuroimaging data. For example, representational similarity analysis and/or encoding/decoding models (Kriegeskorte et al. 2009; Kriegeskorte et al. 2019) provide a direct measure of the representational similarity of models and humans. Another more inexpensive measure that can give a window to internal representations of humans can be eye tracking data, though this may come with its own challenges as well. Another idea could be to decode the agent’s language descriptions, and use alignment techniques such as those in Andreas et al. 2017; Andreas 2019; Wong et al. 2022 Cogsci to see if agent-generated language aligns with program representations or human-generated language. We believe the current work lays a foundation for more systematic probing of internals.
>
> Works cited:
>
> * Kriegeskorte, N. (2009). Relating population-code representations between man, monkey, and computational models. Frontiers in Neuroscience, 35.
>
> * Kriegeskorte, N., & Douglas, P. K. (2019). Interpreting encoding and decoding models. Current opinion in neurobiology, 55, 167-179.
>
> * Andreas, J., Dragan, A., & Klein, D. (2017). Translating neuralese. arXiv preprint arXiv:1704.06960.
>
> * Andreas, Jacob. "Measuring compositionality in representation learning." arXiv preprint arXiv:1902.07181 (2019).
>
> * Wong, C., McCarthy, W. P., Grand, G., Friedman, Y., Tenenbaum, J. B., Andreas, J., ... & Fan, J. E. (2022). Identifying concept libraries from language about object structure. arXiv preprint arXiv:2205.05666.
>
> * Bowers, M., Olausson, T.X., Wong, C., Grand G., Tenenbaum, J.B., Ellis, K., & Solar-Lezama, A. (2022). Top-Down Synthesis For Library Learning. Proc. ACM Program. Lang. 1, CONF, Article 1

---

> ### Author Response · Authors · 2022-08-02
> **Response to Reviewer czvj Part 2**
>
> > Is it possible that the fact that DreamCoder has a DNN inside of it presents a confound that might result in its libraries and procedures being too similar to what an agent would learn directly anyway?
>
> This is a good question. Even though DreamCoder uses a DNN architecture, its architectural modifications, training objective, and data distribution presumably play a critical role in developing representations that can be distinguished from a more ‘vanilla’ DNN that doesn’t necessarily have those three qualities. The DreamCoder DNN’s architecture is built to specifically predict a probability distribution over the current routines in the library (this includes high-level routines added to the DSL during library learning). The DreamCoder DNN is trained to predict the most likely programs for a given grid by balancing a program’s description length and its score (its training objective). In addition, the data it trains on comes both from the GSP task dataset, as well as from grids that are randomly sampled from the current DSL/grammar (“dreams”: its training data). Ellis et al. 2021 shows that this training phase (“dreaming”) has a large effect on enriching the representations of the DreamCoder DNN (see Figs S2 and S3 in the DreamCoder paper).   We conjecture that this combination of architectural priors, more structured training objective, and structured training data allows the program-based representation to serve as a distinct and effective cotraining target that can better instill human inductive biases in more tabula rasa neural agents. We provided a control experiment in the revised manuscript in the supplementary (Figure 8) where we co-trained the RL agent with a more standard ‘vanilla’ DNN. The results show that the selective performance improvement on human-generated tasks vs machine-generated tasks is unique to cotraining representations from the DreamCoder DNN as opposed to a standard DNN. Future work can confirm this by reproducing and extending the experiments of Ellis et al. 2021 in examining and dissecting the representations of the DreamCoder recognition model for our domain. We can also follow this up with more control experiments examining different kinds of DNNs with a variety of architectures, training objectives, and training data and comparing/contrasting how they function as co-training signals for RL agents.
>
> Works cited:
>
> * Ellis, K., Wong, C., Nye, M., Sablé-Meyer, M., Morales, L., Hewitt, L., ... & Tenenbaum, J. B. (2021, June). Dreamcoder: Bootstrapping inductive program synthesis with wake-sleep library learning. In Proceedings of the 42nd acm sigplan international conference on programming language design and implementation (pp. 835-850).

---

### Author Response · Authors · 2022-08-02
**Note to All Reviewers**

We are very grateful to all the reviewers for providing the very helpful and constructive feedback. There were a lot of great points raised about exploring potential avenues for scaling this approach up, exploring the role of abstraction in other kinds of environments, etc. We have included as many main points as we could within the page limit (9 pages) of the submission, but are excited to incorporate more details/points from these discussions in the camera ready version (which gives us 10 pages) should this work be accepted, or in future iterations of this work.

---

### Meta-Review · Area_Chair_3Xv9 · 2022-08-26

**Recommendation:** Accept
**Confidence:** Certain

**Metareview:**

The submission explores differences in human and machine inductive biases. Using the task of generating a pattern in a grid, the authors first show that models trained on human inputs generalize better to machine generated inputs than other human inputs, suggesting that models lack the correct inductive bias. However, by exploiting representations of natural language descriptions or programs during training, models can be given a human-like inductive bias. The reviewers agree that this is creative, thought provoking work backed up by thorough experiments, and the paper it is particularly well written. The main area for improvement in future work would be showing that these results would generalize to other, more complex domains.

**Award:**

Yes

---

### Decision · Program_Chairs · 2022-09-14

Accept